# Breaking with traditions: Who are the women with attitudes, norms and behaviors that support ending female genital mutilation in Burkina Faso?

Ilene S. Speizer[1,2]*, Elizabeth Gummerson[3], Fiacre Bazie[4], Meagan E. Byrne[5], Yentema Onadja[4], Phil Anglewicz[6,7], Georges Guiella[4]

1 Department of Maternal and Child Health, Gillings School of Global Public Health, University of North Carolina at Chapel Hill, Chapel Hill, North Carolina, United States of America, 2 Carolina Population Center, University of North Carolina at Chapel Hill, Chapel Hill, North Carolina, United States of America, 3 Independent Consultant, Baltimore, Maryland, United States of America, 4 Institut Supérieur des Sciences de la Population (ISSP), Université Joseph Ki-Zerbo, Ouagadougou, Burkina Faso, 5 Bureau of Vital Statistics, New York City Department of Health and Mental Hygiene, New York, New York, United States of America, 6 William H. Gates Sr. Institute for Population and Reproductive Health, Johns Hopkins Bloomberg School of Public Health, Baltimore, Maryland, United States of America, 7 Department of Population, Family and Reproductive Health, Johns Hopkins Bloomberg School of Public Health, Baltimore, Maryland, United States of America

* ilene_speizer@unc.edu

## Abstract

Studies examining norms and behaviors around female genital mutilation (FGM) are needed to accelerate progress towards the elimination of this harmful practice. This study uses nationally representative data from a pilot FGM module among women ages 15–49 from Burkina Faso collected in 2023/2024. Three main outcomes related to attitudes, norms and behaviors are used to determine which women who themselves experienced FGM are early adopters of norms and behaviors for stopping the practice in the future. Of the 95% of women who know about FGM, 69.5% experienced FGM, with higher prevalence in rural (74%) than in urban (55%) areas. Results demonstrate that among women who experienced FGM, education level and living in a Christian-headed household are associated with having early adopter attitudes, norms, and behaviors. Conversely, girls and younger women (ages 15–24) are less likely to have early adopter attitudes, norms, and behaviors than their older counterparts (ages 35+). Further, in the analysis of whether women who experienced FGM themselves performed FGM on their daughter(s) or if they would consider FGM for a daughter, for those who do not have daughters, we find that those who are innovators and not practicing FGM are more educated and Christian. In addition, younger women are more likely to report that they would continue FGM than their older counterparts. Results are discussed in the context of strategies to eliminate FGM, particularly among women in communities where FGM is common.

**Data availability statement:** Data for this study are publicly available with no restrictions from this public repository: https://pma.ipums.org/pma/.

**Funding:** This work was conducted with support received from the UNFPA-UNICEF Joint Programme on the Elimination of Female Genital Mutilation (PA) and with support from the Gates Foundation (PA). The UNFPA-UNICEF team gave inputs into the 10 questions on female genital mutilation included in the survey but did not play any role in the conceptualization, analysis, and presentation of the analyses for this paper.

**Competing interests:** The authors have declared that no competing interests exist.

## Introduction

Female genital mutilation (FGM) remains an important global priority as over 230 million girls and women currently alive have undergone the practice, and it is estimated that around four million girls are subjected to the practice each year [1]. The overwhelming majority of FGM cases across the world are in sub-Saharan Africa with prevalence varying from less than 1% in Uganda to 99% in Somalia [1]. Elimination of FGM is one of the targets related to Sustainable Development Goal 5 of achieving gender equality and empowering all women and girls. While some countries have made notable strides in the reduction of FGM [1–3], including Burkina Faso, the site of this study, complete elimination is a challenging endeavor. This is because progress takes time, particularly to change social norms and behaviors related to the practice. At this time, the key players in the field including the United Nations Children's Fund (UNICEF) and the United Nations Population Fund (UNFPA) are calling for the need to accelerate programming progress to attain the 2030 country targets for FGM elimination [1].

A number of behavior change theories have been examined in relation to FGM to inform the readiness for change within a population and how social influences and norms influence the transition away from this common practice [4,5]. Two theories are particularly relevant: Diffusion of Innovations and the Trans-Theoretical Model [4]. Diffusion of Innovations examines the processes that are associated with changes across a population. This theory recognizes that adoption of an innovation is a process starting from innovators (i.e., those who first learn about or create the innovation) to early adopters, to early majority, to late majority, and finally the laggards [6]. In this case, the innovation is the perspective that FGM should be stopped. The Trans-Theoretical Model of behavior change identifies particular stages of change from pre-contemplation, to contemplation, to preparation, to action, to maintenance [7]. In this case, the behavior change is to not circumcise one's daughter. As part of behavior change programming, the innovators and early adopters of a norm or behavior can be engaged to help spread program messages and increase the pace of behavior change.

The classification of FGM as a social norm in communities where it is practiced is based on members considering it an expected behavior and conforming to it based on community expectations and anticipated benefits, as well as perceived consequences of not conforming [5,8–11]. The examination of FGM with a social norms lens provides a strategy for understanding how and why FGM persists even considering global commitments to eliminate the practice [5]. Two particular types of norms are influential here: descriptive and injunctive norms. Descriptive norms around FGM relate to the belief about how common the behavior is. Injunctive norms relate to the extent to which individuals perceive that influential people expect them to practice the behavior (i.e., FGM) [12]. Based on the importance of both injunctive and descriptive norms, it is expected that programs to influence FGM practice need to work with communities and individuals within communities to change the perceived acceptability and implications of not practicing FGM [8].

The UNICEF 2013 seminal report on FGM shows a surprising disconnect between the prevalence of FGM and attitudes toward the continuation of the practice. Across the 29 countries where FGM is practiced and the UNICEF report had data, the majority of girls and women in most countries reported that the practice should stop. This was particularly apparent in Burkina Faso where in 2010, 75% of girls and women had experienced FGM and 90% reported that the practice should be discontinued [8]. There was also widespread support for stopping FGM among men in the Burkina Faso data [8]. Notably, both women and men in Burkina Faso reported no benefit of the practice, but FGM continues in Burkina Faso to this day.

In a qualitative study from Senegal and The Gambia, Shell-Duncan and Hernlund [4] examined attitudes and behaviors around FGM. The authors identified five readiness for change categories among women. They classified women with attitudes in support of stopping FGM and who did not perform (or intend to perform) FGM on their daughters as "willing abandoners." The other extreme was women who think that FGM should continue and have done it (or intend to do it) to their daughters; these women are called "willing adherents." In between these two categories are women who think the practice should be stopped but practiced it (or intend to) on their daughter who are called "reluctant adherents" and women who think it should be continued but did not (or would not) practice it on their daughters who are classified as "reluctant abandoners." Finally, there are the women who are undecided and are classified as "contemplators." Notably this classification scheme is based on attitudes and not norms; however, those in the two reluctant groups are likely influenced by injunctive norms, that is, they are influenced by what they think others expect them to do. Thus, examination of attitudes and injunctive norms jointly can capture another layer of social influences. This is the approach taken in this paper that examines: a) a woman's attitudes in conjunction with what she thinks is expected of her (i.e., injunctive norms); b) a woman's behaviors related to FGM for her daughter(s); and c) a woman's attitudes and behaviors as classified by Shell-Dunca and Hernlund [4].

Using data collected in 2023/2024, we examine attitudes, norms and behaviors that are consistent with the prevention and elimination of FGM. We identify which women would be considered early adopters of abandonment (of FGM) norms and behaviors. We examine the demographic characteristics of the early adopters in Burkina Faso, a country where significant declines have been observed in FGM in the last 30 years from about 80% prevalence in the 1990s to 56% prevalence in 2021 [1]. Information on who the early adopters of abandonment norms and behaviors are can be used to help tailor future interventions that seek to reduce and eliminate FGM in Burkina Faso and other neighboring countries.

## Context

Burkina Faso is a land-locked country in West Africa with an estimated 2025 population of about 24 million inhabitants. The population of Burkina Faso is young with 47% of the population under age 15 years of age; the population is also predominately rural with 72% from rural areas [13]. In Burkina Faso, there has been ongoing legal and policy effort to reduce FGM. A national policy against FGM was passed in 1996 which criminalized FGM being performed by members of the medical community and makes those who know about FGM and do not report it criminally liable [14]. An additional reproductive health law in 2005 was implemented that reached beyond the medical community and outlawed harmful practices, including FGM, more widely [8,14,15]. More recently, in 2018, the anti-FGM statues in the penal code were strengthened to specify jail time from one to ten years and significant fines for conducting FGM, failing to report FGM, as well as punishment for advocating for FGM publicly [16]. Yet, although FGM has long been illegal in Burkina Faso, the practice continues as seen in the 2021 Demographic and Health Survey (DHS) where 56% of women who had heard of FGM had experienced it [13]. The prevalence of FGM is lower among women ages 15–19 (32%) compared to their older counterparts ages 45–49 (83%); suggesting that FGM is on the decline in the country [13]. Further, among girls ages 0–14 in the 2021 DHS survey, only 9.4% had experienced FGM; this percentage was higher if the mother had experienced FGM (13.6%) compared to those where the mother was not circumcised (0.5%). Notably, this includes only those girls who have already experienced FGM and does not account for the fact that some mothers (and fathers) may still intend to

practice FGM on their daughter in the future. Further, given that FGM is illegal in Burkina Faso, and that the government sponsors anti-FGM public information campaigns, it is possible that there is under-reporting of the cases among daughters of respondents [8,9,14,15].

## Materials and methods

### Data

Using Phase 4 Burkina Faso data collected between 5 December 2023 and 28 February 2024 by the Performance Monitoring for Action (PMA) program, which included ten pilot questions on FGM, this study presents the prevalence of FGM and identifies factors associated with innovative FGM norms and behaviors. The ten questions included in this pilot were designed based on the UNFPA-UNICEF ACT Framework that proposes measures of social norms around FGM to be used as part of monitoring and evaluation [17]. Notably, while the ACT Framework includes numerous constructs across the three domains of A (assess what people know, feel, and do and ascertain normative factors), C (context, especially gender and power and social networks and support), and T (track individual and social change), this pilot was limited to 10 questions predominately focused on FGM awareness and status, descriptive norms, and injunctive norms [18].

The PMA data collection strategy uses resident enumerators for data collection; these individuals have been engaged over multiple rounds of data collection. PMA also includes multiple rounds of cross-sectional data as well as longitudinal data collection. For this study, we focus on the data from the Phase 4 cross-sectional sample which is where the FGM module was introduced. For more details on the PMA data collection methods and approaches, see their website that includes countries covered, survey years, survey topics, and relevant indicators and factsheets across the data collection efforts (https://www.pmadata.org/).

The PMA Burkina Faso survey target sample size was determined based on the modern contraceptive prevalence (mCP) among all girls and women, with a 3% margin of error on national estimates and 5% on sub-national estimates. The sample was drawn as a two-stage clustered random sample and is representative at the national level and for urban and rural areas. In Burkina Faso in Phase 4, nationally representative data were collected from 6,089 girls and women aged 15–49 years (response rate: 93.6%), comprising the cross-sectional sample used for the FGM module analysis.

### Ethics statement

The Institut Supérieur des Sciences de la Population (ISSP) at the Université Joseph Ki-Zerbo in collaboration with Johns Hopkins Bloomberg School of Public Health conducted the PMA Phase 4 survey in Burkina Faso. Researchers received ethical approval for conducting the surveys from the Comité d'Ethique pour la Recherche en Santé (Burkina Faso - No. A14-2020) and the Johns Hopkins Bloomberg School of Public Health (IRB No. 12407). All interviewed respondents provided verbal informed consent that was documented electronically. Those participants who were unmarried and ages 15–17 years at the time of the survey were also asked for verbal consent to participate; this is the standard approved practice in Burkina Faso for these types of large-scale household surveys. This analysis uses de-identified public-release data that can be accessed at pmadata.org.

### Analysis sample and variables

The sample for this study is limited first to those women who had ever heard of FGM (n = 5,758; 94.6%). The other factors that reduced the analysis sample were non-response to the other questions about FGM and missing information on the key demographic factors of interest. The variable with the most missing information is religion, which is reported by the household respondent (n = 228 respondents with missing religion, 3.75% of the sample). The unweighted sample includes more urban than rural residents; however, with survey weights, the sample is representative at the national level with three quarters rural and one quarter urban. All descriptive analyses use survey weights to represent the demographics

of the national population of women who have heard of FGM. As shown in Table 1, in the analysis sample (n = 5,498, unweighted), 23.4% live in urban areas, 53.4% have less than primary education and 27.6% have secondary or higher education. The majority (76%) are married or partnered and 77% ever had a live birth. Distinctions are shown by urban and rural areas; not surprising, women in urban areas are more educated and more likely to be unmarried than their rural counterparts. These demographic results are similar to the full PMA cross-sectional sample (not shown). Finally, as shown in Table 2, about 69.5% of women in the analysis sample experienced FGM; this is slightly higher than in the full PMA sample (63.5%) since those women who never heard of FGM are not included in the denominator for this analysis.

The three key outcomes for this analysis are a) attitudes and norms around whether FGM should continue, b) whether a woman has performed or would consider performing FGM on her daughter, and c) readiness to change. First, each

**Table 1. Background characteristics of women with knowledge of FGM surveyed in Burkina Faso, PMA, 2023-2024.**

| Characteristic | All women (n = 5,498†) | Rural (n = 2,198†) | Urban (n = 3,300†) |
|---|---|---|---|
| Age group* | | | |
| 15-24 | 36.71 | 35.58 | 40.39 |
| 25-34 | 29.75 | 29.46 | 30.70 |
| 35+ | 33.54 | 34.96 | 28.91 |
| Education level*** | | | |
| No formal education | 53.37 | 62.02 | 25.09 |
| Primary | 19.03 | 18.25 | 21.58 |
| Secondary or higher | 27.60 | 19.73 | 53.33 |
| Residence | | | |
| Rural | 76.56 | | |
| Urban | 23.44 | | |
| Religion¥ | | | |
| Catholic | 21.74 | 20.02 | 27.33 |
| Protestant | 4.67 | 4.28 | 5.91 |
| Muslim | 67.8 | 68.53 | 65.40 |
| Traditional religion | 5.80 | 7.16 | 1.36 |
| Parity*** | | | |
| None | 23.19 | 19.32 | 35.84 |
| 1-2 | 24.18 | 23.01 | 28.02 |
| 3-4 | 23.67 | 23.77 | 23.35 |
| 5+ | 28.96 | 33.90 | 12.79 |
| Living arrangement*** | | | |
| Single/widowed/divorced | 24.01 | 18.73 | 41.26 |
| Married/living together | 75.99 | 81.27 | 58.74 |
| People in community expect men to make final HH decisions*** | | | |
| Disagree | 12.65 | 11.24 | 17.25 |
| Agree | 87.35 | 88.76 | 82.75 |
| Worked in the last 7 days*** | | | |
| No | 60.92 | 66.12 | 43.93 |
| Yes | 39.08 | 33.88 | 56.07 |

†Unweighted n's shown, weighted n's are 5,278, 4,041, and 1,237, respectively; all percentages are weighted. ¥ Note religion is based on head of household's reported religion. Sample includes those with non-missing on all key variables for analysis of daughter's FGM status (sample reduced by 11 observations for analysis of attitude/norms – n = 5,487).

+p ≤ 0.10; *p ≤ 0.05; **p ≤ 0.01; ***p ≤ 0.001 - indicates significant differences between urban and rural areas.

**Table 2. Percentage of women with knowledge of FGM reporting that FGM should be abandoned or continued based on their own perception and their perception of what the community expects, FGM practice on daughters, and readiness to change, Burkina Faso PMA, 2023-2024.**

| | Total | Rural | Urban | Woman's FGM experience | | Rural women | | Urban women | |
|---|---|---|---|---|---|---|---|---|---|
| | | | | No FGM | Had FGM | No FGM | Had FGM | No FGM | Had FGM |
| Self/Community perceptions: | n=5,487* | n=2,195* | n=3,292* | n=2,105* | n=3,382* | n=616* | n=1,579* | n=1,489* | n=1,803* |
| Continue/Continue | 16.93 | 18.63 | 11.35 | 2.57 | 23.24 | 3.01 | 24.19 | 1.72 | 19.11 |
| Abandon/Continue (early adopter) | 9.01 | 8.97 | 9.15 | 7.62 | 9.63 | 7.75 | 9.41 | 7.38 | 10.57 |
| Continue/Abandon | 6.99 | 7.50 | 5.31 | 3.16 | 8.67 | 4.05 | 8.72 | 1.45 | 8.43 |
| Abandon/Abandon (early adopter) | 67.07 | 64.89 | 74.19 | 86.65 | 58.46 | 85.19 | 57.68 | 89.45 | 61.89 |
| Experienced FGM | n=5,498* | n=2,198* | n=3,300* | | | | | | |
| No | 30.48 | 26.15 | 44.65 | | | | | | |
| Yes | 69.52 | 73.85 | 55.35 | | | | | | |
| Performed or would consider performing FGM on daughter | n=5,498* | n=2,198* | n=3,300* | n=2,108* | n=3,390* | n=615* | n=1,583* | n=1,493* | n=1,807* |
| No (innovators) | 70.25 | 66.93 | 81.10 | 96.05 | 58.94 | 95.73 | 56.74 | 96.65 | 68.56 |
| Yes | 29.75 | 33.07 | 18.90 | 3.95 | 41.06 | 4.29 | 43.26 | 3.35 | 31.44 |
| Five categories of readiness to change† | n=5,488* | n=2,193* | n=3,295* | n=2,105* | n=3,383* | n=614* | n=1,579* | n=1,491* | n=1,804* |
| Willing adherent | 15.98 | 17.47 | 11.13 | 1.88 | 22.17 | 2.30 | 22.84 | 1.08 | 19.24 |
| Reluctant adherent | 11.73 | 13.21 | 6.90 | 2.00 | 16.00 | 1.92 | 17.21 | 2.15 | 10.73 |
| Contemplator | 6.55 | 7.30 | 4.08 | 3.65 | 7.82 | 4.72 | 8.22 | 1.60 | 6.09 |
| Reluctant abandoner (early adopter) | 3.00 | 3.08 | 2.70 | 1.93 | 3.46 | 2.22 | 3.39 | 1.36 | 3.78 |
| Willing abandoner (innovator) | 62.75 | 58.95 | 75.19 | 90.54 | 50.56 | 88.84 | 48.35 | 93.80 | 60.17 |

*unweighted n's, all percentages are weighted; Shaded cells are women who experienced FGM and are main focus of multivariate analyses of innovators. Note, 8% of women with no knowledge of FGM not included in this table. Including these women in FGM prevalence estimates reduces FGM to 63.52%. †Readiness to change based on Shell-Duncan and Hernlund [4].

woman was asked a question about her attitude: "Do you think that FGM should be continued or should it be stopped?" Those women who report stopped (abandon) are coded one and all others (continue or don't know) are coded zero. Second, women were asked an injunctive norm question: "Do you think your community expects you to continue to practice FGM or abandon FGM?" If the woman reported abandon or that FGM does not exist in her community she is coded one; all others (continue or don't know) are coded zero. Because of the high correlation between these two variables (r=0.56), we create a joint measure of her attitude and her perceived community norm. The four categories based on self/community responses are: continue/continue; abandon/continue; continue/abandon; and abandon/abandon (see Table 2). The early adopters of abandonment norms are those who report that they personally favor abandoning FGM and their community favors abandoning it (abandon/abandon). In addition, the women who say abandon and that they perceive their community as continue (abandon/continue) are also considered early adopters. As can be seen in Table 2, most women who did not experience FGM report "abandon/abandon" (greater than 85%). Thus, for the analysis of early adopters, we are focused on who reports abandon among those who experienced FGM; the shaded columns represent the focused analysis sample. Among the women who experienced FGM, nearly 60% report abandon/abandon; about a quarter in rural areas and a fifth in urban areas report continue/continue. There are nearly 10% who report abandon/continue (early adopter response) and who report continue/abandon.

The second outcome of interest is whether the woman reports that her daughter experienced FGM or that she would perform FGM on her daughter if she hasn't yet or has no daughters (see Table 2). All women who report that they would not perform FGM are considered innovators and coded one while all women who report that they have already performed FGM on their daughter or that they would do it are coded zero (non-innovators). Again, as seen above, nearly all women who did not experience FGM would not do it to their daughters (see Table 2). Thus, the innovator analysis focuses on those women who themselves experienced FGM; the analysis sample is shaded. As seen in Table 2, 57% of women who experienced FGM from rural areas and 69% of women from urban areas are innovators in terms of their behaviors or intended FGM behaviors for their daughters.

The third outcome is created based on Shell-Duncan and Hernlund's readiness to change classification [4]. To create this variable (see Table 2), we used information on the woman's attitudes toward whether FGM should continue or be stopped and her behavior of whether she had FGM performed (or would do it) on her daughter. By crossing these two variables, we create a five category readiness to change variable with the following five categories: a) willing adherents (i.e., support continuation and performed or would perform it on their daughter); b) reluctant adherents (i.e., support stopping and performed or would perform on daughter); c) contemplators (i.e., are undecided about attitudes and/or behaviors); d) reluctant abandoners (i.e., support continuation but did not perform or would not perform on their daughter); and e) willing abandoners (support stopping and did not perform and would not perform on daughter). As for the other outcomes, very few women who did not themselves experience FGM would perform it on their daughters, and the overwhelming majority do not support continuation. Therefore, the focus of this analysis is on those who themselves experienced FGM among whom, 22% are willing adherents, 16% are reluctant adherents, 8% are contemplators, 4% are reluctant abandoners, and 50% are willing abandoners. The main difference between urban and rural areas is that in urban areas there are more willing abandoners (60% vs. 48%) and fewer reluctant adherents (11% vs. 17%). For this analysis, we treat the reluctant abandoners as early adopters since they have not performed FGM on their daughters, even if they think the practice should continue. We consider the willing abandoners as the innovators since they have attitudes and behaviors consistent with ending FGM.

### Analysis approach

All descriptive analyses use survey weights for the percentages shown to reflect the distribution of the population at the national level. Unweighted sample sizes are shown; as mentioned earlier, the unweighted sample is disproportionately urban, which is why it is important to use weights when examining the distributions. For the two outcomes that have multiple categories, multinomial logistic regression analyses are performed. For the behavior outcome (performed or would perform FGM on daughter) that only has two categories, logistic regression analyses are performed. All multivariate analyses focus on the women who experienced FGM to determine demographic factors associated with being an early adopter or innovator around FGM attitudes, norms and behaviors. All multivariate analyses use survey weights and adjust for clustering in the sample.

### Results

In the representative sample of women ages 15–49 years from Burkina Faso who have heard of FGM, we find that 69.52% have experienced FGM (see Table 2). The prevalence of FGM is higher in rural (73.85%) than urban (55.35%) areas. Most women experienced FGM between the ages of 1–9 years (55.8% among women in rural areas and 59.5% in urban areas – not shown). Note that these prevalence estimates are higher than the estimates from the 2021 DHS that showed FGM among women ages 15–49 to be 56% overall with 59% in rural areas and 50% in urban areas [13]. Interestingly, the estimates of FGM prevalence among all women are quite similar for women 35 years and older (76.4% in DHS and 79.9% in PMA – See S1 Fig). The PMA estimates are higher among younger women ages 15–24 (38.5% in DHS and 58.8% in PMA). One possible explanation for the higher FGM prevalence among younger women in the PMA data may be

that there is more disclosure of sensitive behavior with the PMA data collection strategy which relies on resident enumerators who establish trust within the communities where they collect data annually over multiple years. More reflections on these distinctions are provided in the discussion section.

Table 3 presents the multinomial logistic regression results of which women are early adopters of abandonment norms based on their own perspective and their perception of the community perspectives (injunctive norm). As mentioned above, the two categories considered to be early adopters are where the woman says abandon but thinks her community thinks it should be continued (abandon/continue) and where she thinks it should be abandoned and thinks her community also feels it should be abandoned (abandon/abandon). The results comparing these two categories to the continue/continue category are shown in columns 2 and 3. Column 1, which is included for comparison, shows the model for women who report they believe FGM should be continued even while they think their communities feel it should be abandoned (continue/abandon), the 'trailers' as opposed to the early adopters. Model 1 for all women who had FGM demonstrates that those women who report abandon/abandon are more likely to have secondary education, be in the oldest (age 35+) age group, and be Christian. Conversely, the youngest women (ages 15–24 and 25–34) and Muslim women are more

**Table 3.** Multinomial logistic regression relative risk ratios (95% CI) of women's reports of whether FGM should be abandoned or continued based on her own perspective/perceived community expectations, among women who had FGM, Burkina Faso PMA.

| Characteristic | Model 1 - All women who had FGM (Self/Community) | | | | | |
|---|---|---|---|---|---|---|
| | Continue/ Abandon | Abandon/ Continue | Abandon/Abandon | | | |
| | vs. | vs. | vs. | | | |
| | Continue/ Continue | Continue/ Continue | Continue/Continue | | | |
| Education level (Ref: No formal education) | | | | | | |
| Primary | 0.82 (0.43-1.58) | 1.07 (0.51-2.22) | 1.26 (0.89-1.78) | | | |
| Secondary or higher | 0.81 (0.36-1.83) | 1.42 (0.99-2.03)+ | 2.33 (1.55-3.51)*** | | | |
| Age group (Ref: 35+) | | | | | | |
| 15-24 | 0.58 (0.31-1.08)+ | 0.33 (0.19-0.56)*** | 0.27 (0.19-0.37)*** | | | |
| 25-34 | 1.33 (0.82-2.13) | 0.81 (0.52-1.26) | 0.58 (0.51-0.67)*** | | | |
| Residence (Ref: Rural) | | | | | | |
| Urban | 1.23 (0.87-1.72) | 1.06 (0.64-1.75) | 1.13 (0.78-1.63) | | | |
| Religion (Ref: Christian)¥ | | | | | | |
| Muslim | 0.65 (0.35-1.21) | 0.46 (0.32-0.66)*** | 0.31 (0.21-0.47)*** | | | |
| Traditional religion | 0.74 (0.30-1.82) | 1.02 (0.51-2.06) | 0.82 (0.49-1.40) | | | |
| Parity (Ref: None) | | | | | | |
| 1-2 | 0.61 (0.20-1.86) | 1.02 (0.41-2.49) | 1.12 (0.69-1.81) | | | |
| 3-4 | 0.51 (0.13-2.05) | 0.61 (0.31-1.23) | 1.05 (0.61-1.79) | | | |
| 5+ | 0.75 (0.24-2.35) | 0.56 (0.24-1.30) | 1.20 (0.54-2.67) | | | |
| Living arrangement | | | | | | |
| Married/living together | 0.82 (0.30-2.27) | 0.66 (0.39-1.10) | 0.82 (0.53-1.27) | | | |
| Disagree (ref) | | | | | | |
| Agree | 0.54 (0.32-0.90)* | 1.02 (0.54-1.93) | 0.75 (0.56-1.01)+ | | | |
| Yes worked | 0.93 (0.65-1.34) | 1.35 (0.75-2.42) | 0.92 (0.68-1.25) | | | |
| Number of observations (unweighted) | n=3,382 | | | | | |

*(Continued)*

| Characteristic | Model 2 - Rural women who had FGM (Self/Community) | | | Model 3 - Urban women who had FGM (Self/Community) | | |
|---|---|---|---|---|---|---|
| | Continue/Abandon | Abandon/Continue | Abandon/Abandon | Continue/Abandon | Abandon/Continue | Abandon/Abandon |
| | vs. | vs. | vs. | vs. | vs. | vs. |
| | Continue/Continue | Continue/Continue | Continue/Continue | Continue/Continue | Continue/Continue | Continue/Continue |
| Education level (Ref: No formal education) | | | | | | |
| Primary | 0.76 (0.32-1.77) | 1.08 (0.40-2.89) | 1.33 (0.83-2.14) | 1.00 (0.66-1.52) | 1.09 (0.50-2.37) | 1.06 (0.89-1.28) |
| Secondary or higher | 0.72 (0.26-1.98) | 1.10 (0.70-1.73) | 2.05 (1.29-3.26)** | 1.11 (0.66-1.85) | 2.86 (1.67-4.90)*** | 3.18 (2.47-4.10)*** |
| Age group (Ref: 35+) | | | | | | |
| 15-24 | 0.63 (0.27-1.46) | 0.29 (0.15-0.58)** | 0.29 (0.20-0.43)*** | 0.70 (0.32-1.51) | 0.45 (0.19-1.07)+ | 0.24 (0.11-0.52)*** |
| 25-34 | 1.47 (0.79-2.73) | 0.79 (0.44-1.44) | 0.59 (0.51-0.69)*** | 1.09 (0.63-1.85) | 0.94 (0.52-1.69) | 0.58 (0.38-0.86)* |
| Residence (Ref: Rural) | | | | | | |
| Urban | na | na | na | na | na | na |
| Religion (Ref: Christian)¥ | | | | | | |
| Muslim | 0.68 (0.30-1.53) | 0.45 (0.28-0.72)** | 0.31 (0.18-0.51)*** | 0.58 (0.30-1.12)+ | 0.46 (0.30-0.72)** | 0.35 (0.23-0.54)*** |
| Traditional religion | 0.70 (0.24-2.04) | 0.98 (0.45-2.16) | 0.77 (0.42-1.41) | 2.59 (0.57-11.69) | 1.73 (1.27-2.36)** | 2.84 (0.48-16.78) |
| Parity (Ref: None) | | | | | | |
| 1-2 | 0.40 (0.15-1.10)+ | 1.04 (0.29-3.73) | 1.15 (0.61-2.18) | 2.35 (0.72-7.62) | 1.01 (0.59-1.73) | 1.03 (0.46-2.32) |
| 3-4 | 0.36 (0.09-1.44) | 0.57 (0.21-1.54) | 1.15 (0.57-2.31) | 1.73 (0.52-5.79) | 0.77 (0.33-1.82) | 0.86 (0.31-2.44) |
| 5+ | 0.62 (0.19-2.01) | 0.50 (0.17-1.49) | 1.42 (0.53-3.82) | 1.27 (0.26-6.12) | 0.80 (0.30-2.10) | 0.60 (0.23-1.60) |
| Living arrangement | | | | | | |
| Single/widowed/divorced (ref) | | | | | | |
| Married/living together | 0.98 (0.29-1.06) | 0.56 (0.27-1.16) | 0.79 (0.43-1.46) | 0.57 (0.23-1.40) | 0.93 (0.53-1.63) | 0.99 (0.54-1.79) |
| Men in community make household decisions | | | | | | |
| Disagree (ref) | | | | | | |
| Agree | 0.54 (0.28-1.06)+ | 1.01 (0.42-2.44) | 0.82 (0.55-1.23) | 0.55 (0.33-0.89)* | 0.96 (0.45-2.04) | 0.54 (0.33-0.87)* |
| Worked in the last 7 days (Ref: No) | | | | | | |
| Yes worked | 0.88 (0.58-1.35) | 1.38 (0.66-2.91) | 0.89 (0.61-1.28) | 1.27 (0.83-1.96) | 1.39 (0.92-2.11) | 1.18 (0.85-1.63) |
| Number of observations (unweighted) | n=1,579 | | | n=1,803 | | |

Note outcome is Self/Community perspective on whether FGM should be abandoned or continued; + p ≤ 0.10; *p ≤ 0.05; **p ≤ 0.01; ***p ≤ 0.001; ¥ Note religion is based on head of household's reported religion. na-not applicable.

likely to report that the practice should be continued and that their communities think it should be continued. In the comparison between abandon/continue and continue/continue, we also find that the youngest women and Muslim women are less likely to be in the early adopter group (i.e., more likely to report continue/continue).

Models 2 and 3 of Table 3 provide the same comparisons for rural (Model 2) and urban (Model 3) women who had undergone FGM. The results for the abandon/abandon compared to continue/continue are the same as the full sample whereby the younger women, Muslim women and the least educated women are more likely to report continue/continue. Further, in urban areas, we see that women who report that they think that most people in their community expect men to make final household decisions are more likely to report continue/continue than have an early adopter response compared to women who do not think men make household decisions. In the comparison of the other early adopter response

(abandon/continue), the results are similar across rural and urban areas with the youngest ages less likely to give the early adopter response than their older counterparts, and in urban areas, women from a traditional religion are significantly more likely to report abandon/continue than continue/continue compared to their Christian counterparts.

Table 4 provides the logistic regression results of whether the woman reports that her daughter had not undergone FGM or that she would not have future daughters undergo the practice (i.e., innovators) versus those who have had or would have their daughters undergo FGM (not innovators). Column 1 is for all women whereas the three columns that follow are focused on those women who themselves experienced FGM. Column 1 shows that those women who themselves experienced FGM are significantly less likely to be innovators than those women who did not experience FGM (OR: 0.07; 95%CI: 0.05-0.11). Further, in the full sample, we also see that younger women and Muslim women have lower odds of being innovators whereas women with secondary education have higher odds of being innovators; no difference is

**Table 4. Logistic regression odds ratios (95% CI) of women's reports of daughter not circumcised and not considering FGM for future daughters (innovators) compared to those who have circumcised or would circumcise future daughters among those who had FGM themselves, Burkina Faso PMA.**

| Characteristic | All women | All women who had FGM | | |
| --- | --- | --- | --- | --- |
| | | All women | Rural | Urban |
| | Innovator - No to FGM for daughter | Innovator - No to FGM for daughter | Innovator - No to FGM for daughter | Innovator - No to FGM for daughter |
| Education level (Ref: No formal education) | | | | |
| Primary | 1.10 (0.80-1.52) | 1.11 (0.82-1.51) | 1.13 (0.76-1.67) | 1.12 (0.86-1.45) |
| Secondary or higher | 2.02 (1.70-2.39)*** | 1.98 (1.61-2.43)*** | 1.74 (1.42-2.13)*** | 2.88 (2.25-3.70)*** |
| Age group (Ref: 35+) | | | | |
| 15-24 | 0.32 (0.16-0.66)** | 0.31 (0.13-0.76)* | 0.35 (0.11-1.08)+ | 0.27 (0.16-0.46)*** |
| 25-34 | 0.63 (0.45-0.87)** | 0.65 (0.46-0.93)* | 0.63 (0.39-1.04)+ | 0.82 (0.56-1.22) |
| Residence (Ref: Rural) | | | | |
| Urban | 1.37 (0.87-2.15) | 1.41 (0.88-2.26) | na | na |
| Religion (Ref: Christian)¥ | | | | |
| Muslim | 0.33 (0.22-0.50)*** | 0.35 (0.23-0.54)*** | 0.36 (0.21-0.60)*** | 0.31 (0.22-0.43)*** |
| Traditional religion | 0.77 (0.57-1.04)+ | 1.10 (0.75-1.62) | 1.06 (0.67-1.70) | ¥¥ |
| Parity (Ref: None) | | | | |
| 1-2 | 1.06 (0.71-1.60) | 1.13 (0.70-1.81) | 1.21 (0.66-2.21) | 0.80 (0.48-1.33) |
| 3-4 | 0.87 (0.42-1.82) | 0.90 (0.42-1.92) | 1.02 (0.38-2.72) | 0.63 (0.33-1.18) |
| 5+ | 0.65 (0.27-1.57) | 0.66 (0.25-1.74) | 0.77 (0.24-2.53) | 0.36 (0.18-0.69)** |
| Living arrangement | | | | |
| Single/widowed/divorced (ref) | | | | |
| Married/living together | 1.04 (0.82-1.33) | 0.99 (0.82-1.21) | 0.94 (0.73-1.22) | 1.23 (0.96-1.56) |
| Men in community make household decisions | | | | |
| Disagree (ref) | | | | |
| Agree | 1.04 (0.78-1.37) | 0.99 (0.74-1.33) | 0.99 (0.68-1.45) | 1.05 (0.73-1.51) |
| Worked in the last 7 days (Ref: No) | | | | |
| Yes worked | 0.88 (0.70-1.09) | 0.88 (0.69-1.12) | 0.84 (0.63-1.13) | 1.15 (0.96-1.39) |
| Mother FGM status (no FGM) | | | | |
| Experienced FGM | 0.07 (0.05-0.11)*** | na | na | na |
| Number of observations (unweighted) | n=5,498 | n=3,390 | n=1,583 | n=1,779 |

+p ≤ 0.10; *p ≤ 0.05; **p ≤ 0.01; ***p ≤ 0.001; Reference group is - Yes to daughter FGM; ¥ Note religion is based on head of household's reported religion; ¥¥ 28 observations with a traditional religion dropped because no variability in outcome (all report innovative outcome). na-not applicable.

found by place of residence. In the analysis among those who experienced FGM, the results are similar whereby younger women and Muslim women have lower odds of being innovators and those with secondary education have higher odds of being innovators.

Table 5 presents the multinomial logistic regression results examining the comparisons across the readiness to change categories for all women (see S1 Table, Models 2 and 3 for rural and urban women, respectively). Not surprisingly, the results are similar to those shown earlier since the categories in this analysis overlap with those in the first two outcomes. In the full sample results, we see that age remains important such that the youngest women (age 15–24 years) are significantly more likely to be willing adherents (performed or would perform FGM on daughter and supportive of continuing) than be willing abandoners or reluctant abandoners (both groups did not or would not perform FGM on daughter) or reluctant adherents (i.e., who have an attitude to stop). In addition, the younger women are less likely to be contemplating (i.e., undecided about attitude or behavior) and more likely to be willing adherents. Some factors related to being early adopters (reluctant abandoners) include being married and living in urban areas. Further, the innovators (support stopping

**Table 5. Multinomial logistic regression relative risk ratios (95% CI) of women's readiness for change based on FGM attitudes and behaviors, Burkina Faso PMA.**

| Characteristic | Model 1 - All women who had FGM | | | | |
| | Reluctant adherent vs. Willing adherent | Reluctant abandoner vs. Willing adherent | Willing abandoner vs. Willing adherent | Contemplator vs. Willing adherent | Reluctant adherent vs. Willing abandoner |
|---|---|---|---|---|---|
| Education level (Ref: No formal education) | | | | | |
| Primary | 1.09 (0.74-1.60) | 0.87 (0.50-1.53) | 1.09 (0.70-1.70) | 0.92 (0.42-2.01) | 1.00 (0.82-1.21) |
| Secondary or higher | 1.36 (1.07-1.72)* | 0.81 (0.37-1.79) | 2.45 (1.90-3.14)*** | 0.98 (0.53-1.80) | 0.56 (0.46-0.68)*** |
| Age group (Ref: 35+) | | | | | |
| 15-24 | 0.28 (0.12-0.64)** | 0.23 (0.11-0.52)*** | 0.19 (0.09-0.39)*** | 0.19 (0.06-0.61)** | 1.46 (0.37-5.79) |
| 25-34 | 0.58 (0.38-0.87)* | 0.68 (0.28-1.64) | 0.46 (0.34-0.61)*** | 0.85 (0.61-1.17) | 1.26 (0.77-2.07) |
| Residence (Ref: Rural) | | | | | |
| Urban | 0.72 (0.45-1.16) | 1.82 (0.97-3.39)+ | 1.24 (0.76-2.02) | 0.84 (0.47-1.51) | 0.58 (0.30-1.15) |
| Religion (Ref: Christian)¥ | | | | | |
| Muslim | 0.85 (0.52-1.38) | 0.64 (0.29-1.41) | 0.31 (0.22-0.44)*** | 1.10 (0.69-1.76) | 2.73 (1.43-5.21)** |
| Traditional religion | 1.17 (0.28-4.79) | 2.43 (0.43-13.66) | 1.24 (0.70-2.18) | 0.73 (0.33-1.63) | 0.94 (0.36-2.45) |
| Parity (Ref: None) | | | | | |
| 1-2 | 0.94 (0.50-1.77) | 0.90 (0.40-2.04) | 1.24 (0.59-2.60) | 0.39 (0.20-0.76)** | 0.76 (0.56-1.04)+ |
| 3-4 | 0.88 (0.52-1.48) | 0.56 (0.20-1.61) | 0.99 (0.40-2.47) | 0.37 (0.12-1.15)+ | 0.89 (0.41-1.93) |
| 5+ | 1.36 (0.56-3.31) | 0.69 (0.23-2.11) | 0.97 (0.29-3.26) | 0.42 (0.10-1.75) | 1.41 (0.50-3.99) |
| Living arrangement | | | | | |
| Single/widowed/divorced (ref) | | | | | |
| Married/living together | 0.86 (0.65-1.14) | 2.30 (1.16-4.55)* | 0.93 (0.71-1.22) | 1.25 (0.77-2.03) | 0.92 (0.71-1.21) |
| Men in community make household decisions | | | | | |
| Disagree (ref) | | | | | |
| Agree | 0.92 (0.65-1.31) | 1.91 (0.67-5.43) | 1.01 (0.81-1.24) | 0.77 (0.34-1.75) | 0.91 (0.62-1.35) |
| Worked in the last 7 days (Ref: No) | | | | | |
| Yes worked | 1.06 (0.62-1.79) | 0.60 (0.38-0.97)* | 0.93 (0.67-1.29) | 0.95 (0.62-1.44) | 1.14 (0.75-1.72) |
| Number of observations (unweighted) | n=3,383 | | | | |

Note outcome is readiness to change category (based on attitude on whether FGM should be abandoned or continued and whether she performed or would perform FGM on daughter); + p ≤ 0.10; *p ≤ 0.05; **p ≤ 0.01; ***p ≤ 0.001; ¥ Note religion is based on head of household's reported religion.

and did/would not perform) compared to the willing adherents have secondary education and practice Christian religion. These findings are similar to those presented earlier with the other outcomes. The models with rural and urban women who experienced FGM are included in the supplementary table (S1 Table) and the results are similar to what was shown earlier such that the rural results are similar to the full sample and the urban results highlight the role of practicing a Christian religion. Further, in urban areas, those women who report that men in the community make household decisions were significantly more likely to be willing adherents than to have attitudes supportive of stopping FGM (reluctant adherents). Interestingly, in urban areas, those who have more children are more likely to be reluctant adherents (i.e., have attitudes that support stopping FGM) but may still be practicing the behavior (e.g., the last column comparing reluctant adherents to willing abandoners). In this case, the women from urban areas may be giving the socially accepted or expected response that they would stop the practice, possibly because they know it is illegal, even if they are still adhering to the practice.

The finding that younger women are less likely to have innovative perspectives is interesting and worth exploring. In Fig 1 and Fig 2, we examine the age differences across the three outcomes for all women (Fig 1) and for women who experienced FGM (Fig 2). Notably, a significantly smaller percentage of women in the youngest ages experienced FGM themselves (Fig 1), which suggests that changes are underway in Burkina Faso. That said, more than half of 15–24-year-olds have still experienced FGM, and these younger women who experienced FGM are significantly less likely to report early adopter or innovative responses than their older counterparts (Fig 2). This is an important area to explore in more depth with qualitative data.

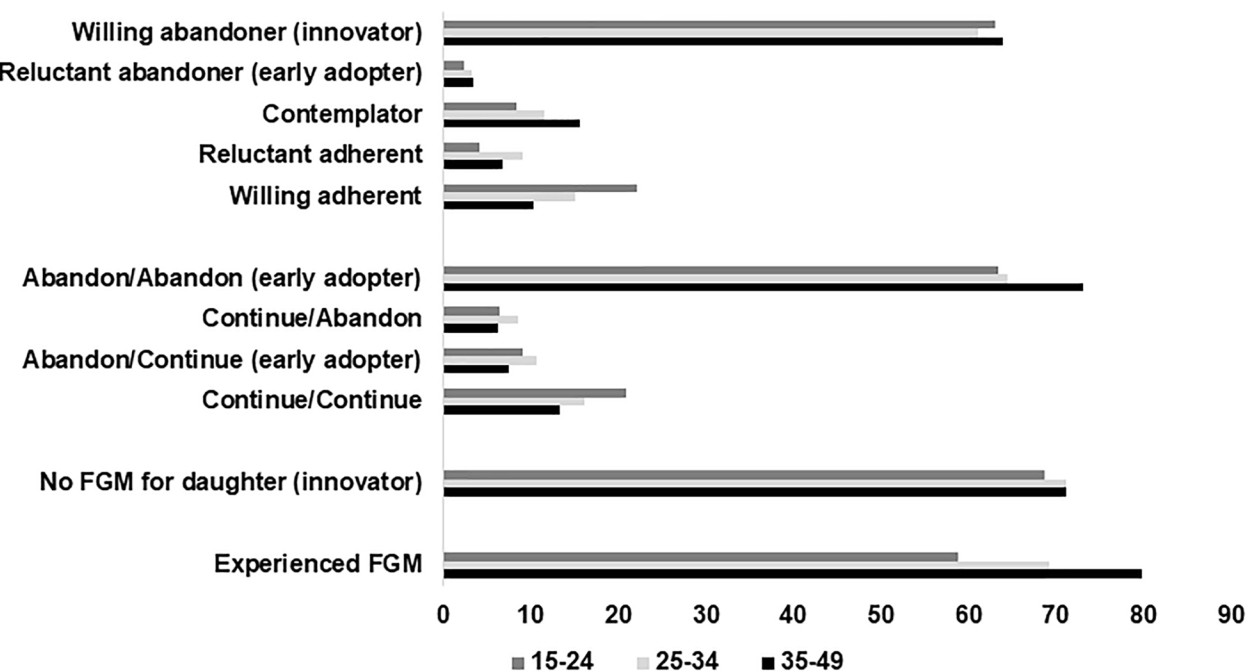

Note: Sample is all women who heard of FGM. Top distribution is readiness to change, middle distribution presents her own attitude and her perspective of her community (self/community) on whether FGM should be continued or abandoned; ***Significant difference for FGM experience by age group at p<0.001

**Fig 1. Percentage of women who have heard of FGM based on the three outcomes of FGM attitudes, norms and behaviors and their own FGM experience, Burkina Faso PMA.**

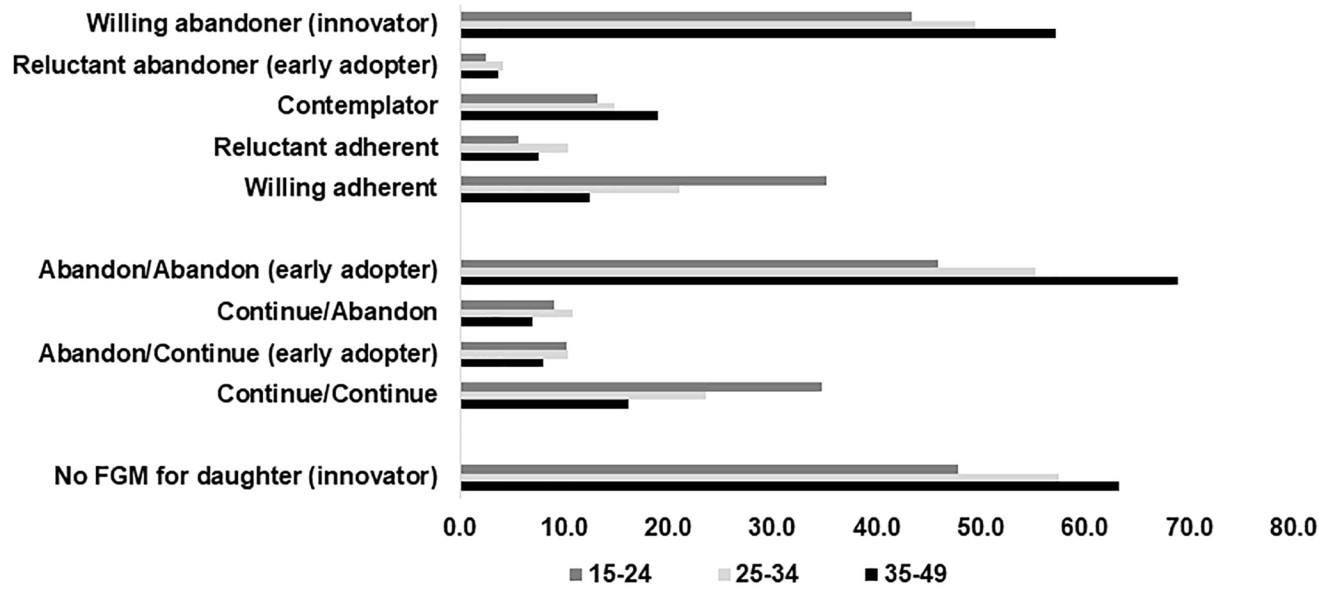

Note: Sample is those women who have heard of FGM and experienced it. op distribution is readiness to change, middle distribution presents her own attitude and her perspective of her community (self/community) on whether FGM should be continued or abandoned; Significant differences for all distributions by age group at p<0.01.

**Fig 2. Percentage of women who experienced FGM based on the three outcomes of FGM attitudes, norms and behaviors, Burkina Faso PMA.**

## Discussion

This study begins to identify who is changing their norms and becoming early adopters of abandonment norms and innovative FGM behaviors in Burkina Faso, especially among women who themselves experienced FGM. While the prevalence of FGM has declined in Burkina Faso from about 80% prevalence in the 1990s to 56% prevalence in 2021 [1,2], the country is not close to elimination, given that many (59% in PMA and 39% in DHS) of the youngest age group (15–24 years) that have heard about FGM have themselves experienced FGM. Education among women is an important protective factor for moving to social norms and behaviors supporting the elimination of FGM. With an increasing emphasis on educational attainment in Burkina Faso [19], it is hypothesized that norms against FGM will continue to change to be more supportive of elimination among those who will become parents in the future and be making decisions about FGM for their daughters. The finding that more educated individuals are more likely to be early adopters of abandonment norms around FGM is seen in other studies that have examined who has FGM perspectives towards elimination [11,20,21]. For example, in a study of two counties in Kenya, those with greater education were more likely to disagree with harmful gender norms related to FGM (and child marriage) [20].

   Norms and behaviors that support the elimination of FGM were found among Christian women and conversely norms supporting the continuation were more common among Muslim women. While FGM is found among both Christian and Muslim communities globally, it has been found that the countries with the highest prevalence of FGM are typically Muslim majority [22]; this is consistent with our findings that those who are Muslim were less likely to have early abandonment norms and innovative behaviors. Thus, programs seeking to reduce and eliminate FGM need to work closely with Muslim communities and Muslim religious leaders in higher prevalence communities in Burkina Faso. Interestingly, among the women who experienced FGM from urban areas, we found that those who were coded as practicing a traditional religion were more likely to be early adopters of abandonment norms than their Christian counterparts; this was an unexpected

result given that FGM is rooted in traditional practices [23]. One possible explanation is that in Burkina Faso, some individuals practice both a mainstream religion (e.g., Christianity or Islam) as well as a traditional religion and thus how they are coded may not be consistent across surveys and households, especially since the religion variable is reported by the head of the household. This is a limitation of how the data were collected.

Of note, we find that younger women who themselves experienced FGM are relatively more likely to have supportive social norms towards continuing FGM and are more likely to report that they would continue this practice with their daughters. This is surprising and requires greater exploration on whether this relates to young women's more frequent exposure to pro-FGM messages compared to older women or alternatively a reduction in abandonment campaigns in recent times because of the COVID-19 epidemic and other competing priorities. Alternatively, it is possible that this is a cohort effect such that older women have greater awareness of legislation and enforcement, they may be more likely to give the "expected" or socially desirable (and legal) response, they may have a greater understanding of the negative reproductive health implications of FGM, or they may be more comfortable going against the expected norm as they become more autonomous decision-makers with age. In this case, the younger women may lack the education, experience, and autonomy to be early adopters.

The age effect with older women having attitudes, norms and behaviors more supportive of stopping FGM than younger women has been found in other studies in Burkina Faso and elsewhere. For example, in an analysis of FGM among adolescents in Burkina Faso, Greis and colleagues [24] find that among the female sample, those adolescents who are in the oldest age group (18+) were more likely to support abandonment of FGM than their younger counterparts (ages 12–13); this corresponds to our findings among the youngest sample. Similarly, in an analysis of the 2005 Ethiopia Demographic and Health Survey data, Masho and Matthews [25] demonstrate that women ages 15–24 were more likely to believe that FGM should continue than their counterparts ages 40–49. It is imperative to explore these age distinctions through qualitative studies to begin to answer the why questions that are not possible to explore with the current data. This type of future qualitative inquiry would be useful for informing programs that work with the youngest Burkinabe women, in and out of school, who will soon become mothers themselves.

This study has strengths and limitations. A strength of this study is the piloting of a set of ten questions in a FGM module to capture attitudes, norms, and behaviors related to FGM. Our findings demonstrate that with well-trained interviewers, it is possible to collect meaningful information on FGM. Further, it is notable that we demonstrate a higher prevalence of FGM among younger women than found in other large survey tools (e.g., the 2021 Demographic and Health Survey [13]). It is not possible to know which estimate is more accurate; however, given that this survey engaged resident enumerators who have been collecting data in the same study areas for a long time, one might expect that the data collected in the PMA survey may be more reliable than other one-off surveys. Moreover, there is evidence that respondents' familiarity with the PMA resident enumerators leads to higher reporting of potentially sensitive outcomes such as infertility, early sexual debut and child mortality [26]. Similarly, a comparison between PMA and DHS on adolescent sexual and reproductive health questions also found higher reporting of sensitive behaviors like adolescent pre-marital sex, although confidence intervals often overlap [27,28]. Further, while the questions on FGM were similarly asked across the two surveys, the DHS included more questions on the type of procedure and who performed the procedure. That said, the coding of FGM was the same across the two surveys based on the initial question on whether the woman had ever heard of FGM and if she experienced any cutting; both of these questions were part of the initial questions in both of the survey sections.

A limitation of this study is that we were only able to include 10 pilot questions and therefore other key areas of the ACT framework and proposed scale questions could not be captured in this study. A future study specifically focused on FGM is needed to test the full list of measures in the comprehensive ACT framework [17]. Another limitation of the study is that the information on religion comes from the household head and includes missing information as well as assumes that all members of a household practice the same religion. With the data available, it is not possible to know how this affects the results. Finally, for this study, the question about whether the woman's community expects her to continue to practice

FGM is open to interpretation by the woman. Some women many be thinking about their geographic community while others may be thinking about their peer network or their religious community, among others. Future surveys may want to ask specific questions about different influential members of a woman's community to better understand how women's beliefs correspond to those in her various networks.

This study begins to answer some questions about social norms and social norms measurement for the FGM community. Notably, a key research question in the FGM research agenda [29] is "What are the valid measures of change in social and gender norms and practices that should be used in the evaluation of FGM interventions?" Our results demonstrate that identifying both individual attitudes as well as injunctive norms and comparing these begins to identify the early adopters of abandonment norms and examining the readiness to change measure can be used to determine if progress is being made to get more women and other members of a community to change their beliefs and practices. These types of analyses can be used to inform whether programs are successfully changing norms and behaviors among those women who are most likely to continue the practice of FGM – that is, those women who experienced FGM themselves.

Our results support the WHO guideline recommendation: "Women and girls living with or at risk of any type of FGM, as well as men and boys from communities that perform FGM, should be provided with educational interventions such as group health education (in health facilities and/or outreach settings, including in humanitarian settings and among refugees), one-on-one FGM education, information sharing or FGM-prevention counselling" [30]. Following from lessons learned on gender transformative interventions, it is crucial that interventions reach both boys and girls (and men and women), incorporate multi-level approaches, and engage multiple sectors [8,29,31]. Thus, programs in Burkina Faso are needed that target girls and women and boys and men in high prevalence communities. Approaches need to tailor messages to Muslim communities and identify how to reach less educated women and girls (and men and boys) who are out of school as well as undertake health education and community dialogues to reach parents, religious and community leaders in higher prevalence communities [31,32]. Given the higher support for FGM among younger women in this and other studies, there is an urgent need for qualitative research to understand the drivers of youth support for FGM. Media campaigns and community outreach that discuss the legality of FGM as well as the health and well-being risks of the practice are needed to potentially fill gaps in programming that may exist in recent years because of COVID-19 and other health priorities. Notably, with availability of global health funding currently in flux, there is a risk of a generation of young people being missed with prevention messaging in support of the global goal of elimination of FGM by 2030. At this juncture, it is essential to implement a comprehensive set of activities within high-risk areas to influence social norms and behaviors around FGM if the future generation of girls are to be protected from this harmful practice.

## Supporting information

**S1 Table. Multinomial logistic regression relative risk ratios (95% CI) of women's readiness for change based on FGM attitudes and behaviors, Burkina Faso PMA, rural (Model 2) and urban (Model 3) women who had FGM.**
(DOCX)

**S1 Fig. Percentage (95% confidence interval) of women who experienced FGM among those who have heard of FGM in Burkina Faso by survey and age group.**
(TIF)

## Acknowledgments

The authors thank Wisal Ahmed, Souhi Paulin Fidèle Tra, Menbere Legesse, and Berhanu Legesse of the UNFPA-UNICEF Joint Programme on the Elimination of Female Genital Mutilation for their contributions to the design and conceptualization of the pilot FGM module.

## Author contributions

**Conceptualization:** Ilene S. Speizer, Elizabeth Gummerson.

**Data curation:** Fiacre Bazie, Georges Guiella.

**Formal analysis:** Ilene S. Speizer.

**Funding acquisition:** Phil Anglewicz.

**Investigation:** Ilene S. Speizer, Elizabeth Gummerson, Fiacre Bazie, Meagan E. Byrne, Yentema Onadja, Phil Anglewicz, Georges Guiella.

**Methodology:** Ilene S. Speizer.

**Project administration:** Fiacre Bazie, Meagan E. Byrne, Phil Anglewicz.

**Supervision:** Georges Guiella.

**Validation:** Fiacre Bazie, Yentema Onadja, Georges Guiella.

**Writing – original draft:** Ilene S. Speizer.

**Writing – review & editing:** Elizabeth Gummerson, Fiacre Bazie, Meagan E. Byrne, Yentema Onadja, Phil Anglewicz, Georges Guiella.

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
