## [Decision Letter · Decision Letter 0]

11 Sep 2025

PGPH-D-25-01698

Breaking with Traditions: Who Are the Innovators that Support Ending Female Genital Mutilation in Burkina Faso

Dear Dr. Speizer,

Thank you for submitting your manuscript to PLOS Global Public Health. After careful consideration, we feel that it has merit but does not fully meet PLOS Global Public Health’s publication criteria as it currently stands. Therefore, we invite you to submit a revised version of the manuscript that addresses the points raised during the review process.

The manuscript has been evaluated by two reviewers, and their comments are available below. 

The reviewers have raised a number of major concerns. They feel the manuscript should outline a clearly-defined research question, and they request improvements to the reporting of methodological aspects of the study.

Could you please carefully revise the manuscript to address all comments raised?

We look forward to receiving your revised manuscript.

Kind regards,

Jennifer Tucker, PhD

Staff Editor

Journal Requirements:

If you did not receive any funding for this study, please simply state: “The authors received no specific funding for this work.

Additional Editor Comments (if provided):

Reviewer #1:

Reviewer #2:

Reviewers' comments:

Reviewer's Responses to Questions

**Comments to the Author**

1. Does this manuscript meet PLOS Global Public Health’s publication criteria?

Reviewer #1: Yes

Reviewer #2: Yes

2. Has the statistical analysis been performed appropriately and rigorously?

Reviewer #1: Yes

Reviewer #2: Yes

3. Have the authors made all data underlying the findings in their manuscript fully available (please refer to the Data Availability Statement at the start of the manuscript PDF file)?

Reviewer #1: Yes

Reviewer #2: Yes

4. Is the manuscript presented in an intelligible fashion and written in standard English?

Reviewer #1: Yes

Reviewer #2: Yes

Reviewer #1: This manuscript examines demographic factors associated with supportive attitudes, norms, and behaviours toward abandoning female genital mutilation (FGM) among women in Burkina Faso, drawing on recently collected, nationally representative PMA data. The research addresses a highly relevant and policy-critical issue and offers potentially actionable insights for program design. The paper is well-written and clearly structured. Nonetheless, the manuscript would benefit from a more Burkina Faso–specific theoretical framing, a deeper conceptualisation and justification of the term and definition of “innovators,” and a more explicit integration of existing evidence—particularly from UNICEF’s 2013 work—on legislation, FGM support, and underreporting in Burkina Faso.

Major Comments

The introduction currently draws heavily on general or multi-country literature. It should be strengthened by presenting a Burkina Faso–specific theoretical framework. In particular, the seminal UNICEF (2013) study Female Genital Mutilation/Cutting: A statistical overview and exploration of the dynamics of change contains country-level insights on the role of legislation, enforcement, and the underreporting of supportive attitudes due to social desirability bias in survey contexts. These findings could provide essential context for interpreting your results, particularly the unexpectedly high proportion of younger women with pro-continuation views.

Moreover, the current operational definition of “innovators”—based on a combination of personal attitudes and perceived community norms—feels primarily descriptive. I recommend developing a theoretically grounded definition, explicitly linking it to social norms theory (e.g., injunctive/descriptive norms) and the diffusion of innovations framework. This would provide stronger conceptual underpinnings for your classification and ensure it reflects behavioural science theory rather than the convenience of available data. The definition should be robustly grounded in theory. While the term “innovators” is borrowed from the diffusion of innovations theory (which you could explain in more depth), its use here should be discussed and justified more carefully. After the proposed theoretical reframing, you might consider alternative terminology (e.g., “early adopters of abandonment norms” or “norm-changers”) that better captures the social positioning outside the prevailing norm, without unintended value judgements. The choice of term should follow from the conceptual discussion in the introduction.

The finding that younger women with FGM are more likely to support continuation than older women is counterintuitive and warrants deeper theorisation (although it is not entirely new—see Masho and Matthews 2009 for Ethiopia or Ortensi et al. 2025 among migrants in Italy for similar results). Beyond the explanations currently offered, consider, among others:

- Cohort effects linked to the timing of legislation and enforcement in Burkina Faso.

Potential underreporting of support for continuation among older women due to greater awareness of socially acceptable responses.

- Differences in exposure to abandonment campaigns across age cohorts, including possible effects of COVID-19 on the discontinuation or scaling back of prevention campaigns.

The difference between PMA and DHS prevalence estimates is noted but could be unpacked further, considering potential methodological drivers such as interviewer training, trust-building via resident enumerators, and question ordering or phrasing effects.

Reviewer #2: This is an interesting study addressing an important research topic and significant public health issue.

Comments:

1. The title should explicitly include the word “women”, as the term “innovators” could also be interpreted to include men.

2. In lines 78–82, remove the word “Recent” since it refers to the year 2021, which is no longer very recent.

3. In lines 111–113 (Phase 4 Data of PMA), provide more information about this survey and include a reference if available.

4. In line 125, replace the hyperlink with a formal reference.

5. Clarify whether you obtained approval to access and use the survey data. If so, please state this explicitly.

6. In Table 1, it would be more informative to present both numbers and percentages.

7. In Table 1, for the education level variable, use “No formal education” instead of “None.”

8. In Table 1, the percentages for Urban and Rural appear incorrect (should be 40% and 60% instead of 27% and 76%). Please verify these figures. Also, since Urban/Rural are already shown as columns, there is no need to repeat them in the rows.

9. In the Materials and Methods section, consider reorganizing it into standard, clearly labeled subsections for better clarity and flow.

10. Provide more detail on the statistical analysis in the Materials and Methods section.

11. The study relies on only 18 references, many of which are organizational reports rather than peer-reviewed academic papers. Given the importance and the well-researched nature of the topic, more literature—especially academic sources—should be incorporated, particularly in the Discussion.

12. In the Discussion, expand on the implications of your findings in greater depth, addressing practice, policy, and future research aspects.

**Do you want your identity to be public for this peer review?** For information about this choice, including consent withdrawal, please see our Privacy Policy

Reviewer #1: No

Reviewer #2: No

---

## [Decision Letter · Decision Letter 1]

20 Nov 2025

Breaking with Traditions: Who Are the Women with Attitudes, Norms and Behaviors that Support Ending Female Genital Mutilation in Burkina Faso?

PGPH-D-25-01698R1

Dear Dr. Speizer,

We are pleased to inform you that your manuscript 'Breaking with Traditions: Who Are the Women with Attitudes, Norms and Behaviors that Support Ending Female Genital Mutilation in Burkina Faso?' has been provisionally accepted for publication in PLOS Global Public Health.

Best regards,

Julia Robinson

Executive Editor

Reviewer Comments (if any, and for reference):

Reviewer's Responses to Questions

**Comments to the Author**

Reviewer #1: All comments have been addressed

Reviewer #2: All comments have been addressed

publication criteria?

Reviewer #1: Yes

Reviewer #2: Yes

3. Has the statistical analysis been performed appropriately and rigorously?

Reviewer #1: Yes

Reviewer #2: Yes

4. Have the authors made all data underlying the findings in their manuscript fully available (please refer to the Data Availability Statement at the start of the manuscript PDF file)?

Reviewer #1: Yes

Reviewer #2: Yes

5. Is the manuscript presented in an intelligible fashion and written in standard English?

Reviewer #1: Yes

Reviewer #2: Yes

Reviewer #1: All comments have been effectively addressed.

Reviewer #2: Thanks for considering my comments.

**Do you want your identity to be public for this peer review?** For information about this choice, including consent withdrawal, please see our Privacy Policy

Reviewer #1: No

Reviewer #2: No
